# Determination of a New Biomarker at the Level of Gene Alteration in Cisplatin Ototoxicity

**DOI:** 10.3390/ijms26188880

**Published:** 2025-09-12

**Authors:** Deniz Kızmazoğlu, Aylin Erol, Tekincan Çağrı Aktaş, Yüksel Olgun, Ayşe Banu Demir, Zekiye Altun, Safiye Aktaş, Nur Olgun

**Affiliations:** 1Department of Pediatric Oncology, Institute of Oncology, Dokuz Eylül University, Izmir 35340, Turkey; 2Department of Basic Oncology, Institute of Oncology, Dokuz Eylül University, Izmir 35340, Turkey; 3Department of Otorhinolaryngology, Faculty of Medicine, Dokuz Eylül University, Izmir 35340, Turkey; 4Department of Medical Biology, Faculty of Medicine, Izmir University of Economics, Izmir 35330, Turkey; 5Acıbadem Kent Hospital, Karsıyaka, Izmir 35630, Turkey

**Keywords:** pediatric cancer, cisplatin ototoxicity, hearing loss, biomarkers

## Abstract

Cisplatin is an alkylating chemotherapeutic drug used in the treatment of many pediatric solid tumors, and cisplatin ototoxicity is characterized by sensorineural, bilateral, irreversible, and progressive hearing loss. The aim of this study is to identify biomarkers that may serve as predictors of cisplatin-induced ototoxicity in pediatric cancers. In our preliminary study, patients with severe hearing loss were analyzed using the comparative genomic hybridization (CGH) method. Mutations were identified in the following genes: *ADAM6*, *SIX3*, *GNAS*, *NDUFV1*, *H19*, *DEFA4*, and *ZIM2*. Based on these data, we aimed to investigate the mutation status of these candidate genes in a larger population of pediatric cancer patients treated with cisplatin. DNA samples were extracted from the mononuclear cells of peripheral blood samples obtained from 82 patients. These genes were analyzed using the RT-PCR technique, and ototoxicity was assessed using the Brock and Muenster classifications. Hearing loss was detected in 28% of patients; 76.8% and 23.2% had mild and severe hearing loss, respectively. A significant correlation was found between *ZIM2* gene amplification and the presence of ototoxicity (rho = 0.461, *p* = 0.003), especially in advanced-stage cancer patients with severe hearing loss (rho = 0.38, *p* = 0.017). Our findings suggest that *ZIM2* is a promising biomarker for predicting cisplatin ototoxicity.

## 1. Introduction

Platinum-based agents such as cisplatin, oxaliplatin, and carboplatin are used in pediatric cancer treatment [1]. Examples of these cancer types include central nervous system tumors, neuroblastoma, hepatoblastoma, nasopharyngeal cancer, retinoblastoma, soft tissue sarcomas, and germ cell tumors [2]. However, cisplatin has serious side effects, including ototoxicity, neurotoxicity, and nephrotoxicity. As is known, platinum-induced ototoxicity occurs in 42–67% of patients receiving cisplatin treatment. If severe ototoxicity occurs during this treatment, the dose of cisplatin may be reduced, or cisplatin treatment may be discontinued to prevent ototoxicity development. However, these attempts may decrease the antitumor effects of the drug [3]. Epidemiological studies have identified various risk factors that depend on the type and dose of the platinum-based chemotherapeutic agent [4]. Accordingly, studies have been performed to determine genetic predis-position by focusing on gene variants that cause changes in the pharmacokinetics and pharmacodynamics of cisplatin [5,6].

Both candidate gene analyses and genome-wide association studies (GWASs) have explored the genetic contribution to platinum-based-agent-induced ototoxicity in pediatric cancer patients. These investigations have identified significant associations between cisplatin-induced ototoxicity and genetic variants in several genes, most notably thiopurine methyltransferase (*TPMT*), catechol-O-methyltransferase (*COMT*), glutathione S-transferase pi 1 (*GSTP1*), and superoxide dismutase 2 (*SOD2*) [7,8,9,10,11,12].

Genetic alterations are being studied in the process of identifying new genetic markers and, currently, single-nucleotide polymorphism studies are being conducted to investigate and identify genetic variants in individual patients [5,13]. In various studies cited in the literature, DNA has been extracted from the mononuclear cells of pediatric patients treated with cisplatin and suffering from different levels of hearing loss, and the single-nucleotide polymorphisms of genes that are thought to be or have been shown to be related to cisplatin ototoxicity such as megalin, glutathione S-transferases, *TMPT1*, *COMP1*, and *ERCC1* have been screened. The *SIX3* gene, known to have an important role in forebrain development, is also thought to have a role in cisplatin ototoxicity [14,15]. In our previous studies, polymorphisms in *ERCC1*, *GSTP1*, *LRP2*, *TPMT*, and *COMT* genes were investigated in cases where cisplatin was administered. Although the mutant genotype of *GSTP1* was found to be significantly associated with cisplatin ototoxicity in univariate analysis, multivariate analysis did not confirm this finding [16,17]. Tserga et al. conducted a meta-analysis in their systematic review and found *ACYP2* and *LRP2* genes to be related to antioxidants, neurotransmission, or auditory function associated with cisplatin ototoxicity [18].

Pediatric cancers are relatively rare (between 110 and 150 per million under the age of 15 years) and are classified as lymphoproliferative tumors (leukemia and lymphoma) and solid tumors. With the current use of next-generation sequencing (NGS) in pediatric solid tumors, somatic mutations in these have started to be identified [19,20]. Germline mutations in a wide range of genes predisposing to carcinogenesis appear to have a greater role in the development of pediatric cancers than previously thought [21]. Thanks to the recently performed large-scale genomic analyses, the understanding of genome information and genetic factors in pediatric cancers has improved, helping define new clinically significant subtypes [22,23]. In many pediatric tumors, the impact of the initiating factor depends on the developmental stage at which the tumor arises [20].

Attempts have been made to reveal the bases of genomic alterations in pediatric cancers due to both somatic and germline mutations using an integrative analysis approach beyond DNA panels. In the early stages of sequencing studies performed in pediatric cancers, although the aim was to identify genes acting as oncogenic drivers, the presence of a generally low mutational burden was demonstrated. Surprisingly, even in most of the high-risk and aggressive cancers, an identifiable driver gene or pathway has not been fully revealed. In pediatric tumors, changes in PR Set Region 6 (*PRDM-6*) in a subset of medulloblastomas (MBLs) and in the Telomerase Reverse Transcriptase (*TERT*) gene in NB were detected via sequencing analysis [22].

In this study, the role of genes that may be biomarkers of cisplatin ototoxicity was evaluated. Accordingly, we aimed to discover biomarker(s) that may play a role in the prediction of drug-induced hearing loss and to re-evaluate treatment strategies in light of the new information we have obtained.

## 2. Results

### 2.1. Preliminary Study Results

Our preliminary study included five patients (three males and two females) with hearing loss (grade 2 or higher) according to the Brock and Muenster classifications. These cases were analyzed using comparative genomic hybridization (CGH), which revealed various gene losses and gains. The mean age of the patients was 66 months, and the mean cumulative cisplatin dose was 391 mg/m^2^. Brock scores were 2, 2, 2, 3, and 4, while Muenster scores were 3c, 3a, 3c, 3b, and 2a, respectively. No statistically significant associations were observed between the selected gene polymorphisms and the severity of hearing loss. Genes altered in at least three of the five patients—*ADAM6*, *SIX3*, *GNAS*, *NDUFV1*, *H19*, *DEFA4*, and *ZIM2*—were selected for further investigation in this study.

### 2.2. Results of 82 Analyzed Cases with Seven Candidate Genes

In the main analysis, 82 patients were evaluated for seven candidate genes (*ADAM6*, *SIX3*, *GNAS*, *NDUFV1*, *H19*, *DEFA4*, and *ZIM2*). The male-to-female ratio was 1.15, with 44 males and 38 females. The distribution of tumor diagnoses is presented in Table 1.

The median age at diagnosis was 50 months, ranging from 2 to 210 months. Neuroblastoma was the most common diagnosis, with a mean age at diagnosis of 29 months (range of 2–189 months). The mean cumulative cisplatin dose was 400 mg/m^2^ (range 75–800 mg/m^2^), and the distribution of cisplatin doses across tumor subtypes is illustrated in Figure 1. None of the patients showed hearing loss in baseline audiological tests. Audiological assessments were repeated before each chemotherapy cycle. While treatment continued in patients with grade 1 or 2 hearing loss, it was modified in patients with grade 3 or higher hearing loss.

Ototoxicity was assessed using the Brock and Muenster classifications. Overall, 28% of patients developed ototoxicity during treatment, with 76.8% and 23.2% of these cases classified as mild and severe hearing loss, respectively. The clinical characteristics of patients who developed hearing loss are presented in Table 2.

### 2.3. Gene Alterations and Correlation with Ototoxicity

No amplification or deletion among the seven studied genes was found to be an independent determinant of ototoxicity or hearing loss in pediatric cancer patients treated with cisplatin; this distribution is given in Table 3. *ZIM2* gene amplification was detected in 10 cases, 7 of which were neuroblastoma patients. Spearman correlation analysis revealed a significant association between ototoxicity and *ZIM2* gene amplification in advanced-stage cases (rho = 0.461, *p* = 0.003). Severe hearing loss was particularly correlated with *ZIM2* amplification (rho = 0.38, *p* = 0.017). The 95% confidence interval and effect sizes were manually calculated using the eta squared (η^2^) formula, yielding an effect size of 0.1029, which corresponds to a medium-to-high effect. No statistically significant correlation was found between ototoxicity and *ZIM2* gene deletion (*p* > 0.05). In non-parametric univariate chi-square testing, *ZIM2* amplification was statistically significant for hearing loss (*p* = 0.048, Chi^2^). However, in multivariate analyses (general linear model), *ZIM2* mutation was not an independent factor (*p* = 0.132), and sex, cisplatin dose, and age were not related to any gene mutations.

Most cases were neuroblastoma, with a mean age of 38.66 months (range 2–189). Among these cases, *ZIM2* amplification was observed in 22% of 32 patients and was statistically significant for hearing loss in this subgroup (*p* = 0.044, Chi^2^). Severe ototoxicity was distributed across different tumor groups. No statistically significant correlations were identified between ototoxicity and the amplification or deletion of *SIX3*, *ADAM6*, *DEFA4*, *H19*, *NDUFV1*, or *GNAS* genes (*p* > 0.05).

## 3. Discussion

In this study, we investigated the germline mutation alterations of *ADAM6*, *SIX3*, *GNAS*, *NDUFV1*, *H19*, *DEFA4*, and *ZIM2* genes in 82 pediatric cancer patients treated with cisplatin. These genes were detected in five patients with severe hearing loss by comparative genomic hybridization (CGH) in our preliminary study about cisplatin ototoxicity. These genes were then selected as biomarker candidates. Molecular findings, as well as clinical and audiological follow-up findings related to the ototoxicity status of the patients, were statistically compared with Brock and Muenster classifications. Statistically significant changes were found at the level of amplification in the *ZIM2* gene.

*SIX3* encodes a protein involved in cell development, division, and apoptosis and is part of the *SIX* protein family (*SIX1–6*), which plays a role in inner ear development and prevents auditory system malformations. Mutations in *SIX3* are associated with developmental disorders such as anophthalmia and hypopituitarism [24]. *GNAS* encodes a G protein subunit essential for intercellular signaling and protein communication. Although its role in ototoxicity is not fully understood, postzygotic activating mutations in *GNAS* (20q13.3) have been linked to hearing impairments [25,26]. *NDUFV1* encodes a subunit of mitochondrial complex I, located on chromosome 11q13.2, and plays a crucial role in cellular respiration. By interacting with *NDUFS1*, the largest subunit of complex I, *NDUFV1* stabilizes and enhances the function of the complex. Mutations in *NDUFV1* can impair mitochondrial respiration, contributing to diseases such as Leigh syndrome, which may involve hearing loss [27]. *H19* is an imprinted long non-coding RNA (lncRNA) gene located on chromosome 11p15.5, expressed from the maternal allele. It regulates growth factor signaling pathways that control cell proliferation, differentiation, and development. Altered *H19* expression has been observed in various cancers, affecting tumor growth, metastasis, and therapy resistance [28]. *DEFA4*, a member of the beta-defensin family located on chromosome 20p13, encodes antimicrobial peptides that play a key role in innate immunity. This gene is expressed in multiple tissues, particularly epithelial cells, and increased DEFA4 protein expression has been observed in patients with hearing loss [29]. *ZIM2* is a gene located within the PEG3-imprinted domain and exhibits lineage-specific imprinting patterns in mammals. It functions as a transcription regulator and is part of a family of zinc-finger proteins involved in development and gene expression control. *ZIM2* is associated with multiple cellular functions, including intracellular signal transduction, protein and lipid metabolism, and regulation of calcium transport channels in the endoplasmic reticulum. *PEG3* is a paternally expressed gene related to the maternally expressed *ZIM1*, *ZIM2*, and *ZIM3* genes, with most open reading frames being lost during mammalian development except for *PEG3*. Imprinting control regions (ICRs) regulate these processes, and *ZIM2*, along with related genes, shows increased expression in the brain, testis, nasal area, and accessory olfactory organs. Large-scale genomic analyses in pancreatic cancer have identified *ZIM2* mutations, suggesting a potential role in tumorigenesis. Although the role of *ZIM2* in pediatric cancers remains largely unexplored, its imprinting and involvement in fundamental developmental pathways indicate that further investigation in pediatric contexts is warranted. Mutation-related changes in *ZIM2* imprinting may lead to increased *ZIM2* activity [30,31,32,33].

Our observation of the increase in *ZIM2* activity regarding cisplatin ototoxicity and hearing loss needs further research. In our literature review, we could not encounter any study demonstrating the relationship between the *ZIM2* gene, hearing acuity, and ototoxicity. The impact of *ZIM2* activity on calcium channels may affect drug transport into the inner ear.

Cisplatin may cause sensorineural hearing loss by damaging the stria vascularis layer, degenerating the spiral ganglion, and resulting in loss of hair cells. The three main issues of importance are the effect of reactive oxygen radicals, the molecular mechanisms mediating loss of inner ear cells, and the role of transport channels that transport cisplatin into the inner ear cells. Once inside the hair cells, cisplatin disrupts organelles, affects metabolism, induces oxidative stress, and targets DNA to cause intracellular damage [34]. We think that the effect of *ZIM2* gene amplification may have caused a greater amount of cisplatin uptake into the inner ear cells in these individuals by acting on transport channels due to its known effect on calcium metabolism.

The blood–labyrinth barrier in the inner ear prevents the passage of toxins into the inner ear. The integrity of the blood–labyrinth barrier is related to the integrity of the stria vascularis layer, which is located on the lateral wall of the cochlea and expresses multiple ion transporters. If the marginal layer is underdeveloped, large amounts of cisplatin can enter the inner ear [19].

Anti-inflammatory and immunity-based mechanisms and many receptors, cytokines, chemokines, and proinflammatory signaling molecules play a role especially in eliminating the toxic effects of cisplatin and repairing the resulting damage. Our hypothesis and findings in this study indicate that the level of damage will change with increased or decreased activities due to individual differences in cisplatin ototoxicity and genetic alterations. Therefore, knowing these differences in advance may lead to changes in the planning of treatment with cisplatin or minimize the damage incurred with closer audiological follow-up. These genetic differences may cause ion channels to function differently, resulting in different doses of cisplatin entering the inner ear or the inability to refrain from the toxic effects of the drug.

The limitations of this study are that the tumor diagnosis group was not homogeneous, and the total dose of cisplatin used for each patient was not the same. However, the lowest dose is sufficient to induce ototoxicity in susceptible individuals. Additionally, the relatively small sample size (82 patients) might have limited statistical power, particularly for subgroup analyses based on tumor type. Another limitation is the lack of control groups consisting of cisplatin-treated patients who did not develop ototoxicity and healthy individuals; however, ethical restrictions prevent invasive blood sampling for such studies. Furthermore, since the study was conducted at a single center, it might be told that with patients from a specific regional and ethnic background, the generalizability of the findings to more diverse populations is limited. In fact, our center accepts patients from all around Turkey and regional countries. A strength of this study is that we worked with a cohort of patients whose clinical status was regularly followed and who underwent regular audiological assessments.

This restricts our ability to determine whether *ZIM2* amplification is a specific marker for ototoxicity or represents a broader genomic alteration among pediatric cancer patients. Future prospective studies with larger and more homogeneous cohorts, the inclusion of appropriate control groups, and the representation of diverse populations are needed to validate our findings and further clarify the role of *ZIM2* in cisplatin-induced ototoxicity. Additionally, a lack of in vitro and in vivo functional validation of *ZIM2* limits mechanistic insights, and this will be addressed in future studies.

The technology for discovering cancer biomarkers is evolving every day. For example, the use of artificial intelligence algorithms can help identify cancer biomarkers more accurately. Similarly, single-cell RNA sequencing technology can be used to define molecular characteristics of cancer cells in more detail. In conclusion, the discovery and development of cancer biomarkers will continue to provide a greater number of alternatives with the potential for more accurate cancer diagnoses, better curative treatments, and improved prognoses.

Studies on the relationship between cancer biomarkers and ototoxicity will help in developing a safer, effective, and individualized approach to cancer treatment. Our findings suggest that *ZIM2* gene amplification may increase sensitivity to cisplatin ototoxicity. Detailed investigation into the relationship of the *ZIM2* gene with ear tissues, hearing acuity, and drug metabolism has emerged as a future goal.

## 4. Materials and Methods

### 4.1. Preliminary Study

Five patients with severe hearing loss after cisplatin treatment were selected for the preliminary study. Comparative genomic hybridization (CGH) analyses were performed using NimbleGen CGH v2 (Roche NimbleGen, Inc., Madison, WI, USA). of all chromosomes were performed on DNA extracted from peripheral blood mononuclear cells, and this DNA was isolated using the spin column method. DNA from the five cases (test DNA) and normal DNA (control DNA, Roche standard DNA) were labeled with different fluorochromes and hybridized with metaphase chromosomes obtained from normal cells.

Differences in fluorescence intensities along chromosomes in the reference metaphase area indicate copy number changes (amplifications or deletions) in tumor DNA. DNAs showing different fluorescence phenotypes appear in blue-orange if they overlap when hybridized with normal metaphase chromosomes. If there is a deletion in the test DNA, hybridization does not occur in that region, so it appears red. If there is an amplification in the DNA, hybridization is more intense, and the region appears brighter green. CGH profiles were calculated on the computer using Agilent CytoGenomics Software, version 2.0 (Agilent Technologies Inc., Santa Clara, CA, USA). We found through CGH analysis that the *ADAM6*, *SIX3*, *GNAS*, *NDUFV1*, *H19*, *DEFA4*, and *ZIM2* genes were mutated in five patients with severe hearing loss [35]. In line with these data, we aimed to detect the proliferation of these genes in a larger patient population.

Patients aged 0–18 years were included in this study. Only those with audiological assessments available at baseline and during follow-up after treatment were considered. All cases underwent successful DNA extraction and were included in the PCR analysis. Patients exhibiting signs of external or middle ear disease and/or with pre-existing hearing loss were excluded from the study. This study did not include control cases, because the aim was to find a molecular biomarker to predict patients who were more sensitive to cisplatin and more likely to develop ototoxicity. All patients included in this cohort received cisplatin chemotherapy in the Pediatric Oncology Department of Dokuz Eylül University.

### 4.2. Informed Consent

Parental informed consent was obtained for all participants who received cisplatin treatment during routine follow-up visits at the Department of Pediatric Oncology, Dokuz Eylül University.

### 4.3. Ethics Committee Approval

This study was approved by the Non-Interventional Research Ethics Committee of Dokuz Eylül University (Approval No: 2022/39-13, Date: 7 December 2022).

### 4.4. Collection of Samples

In addition to routine control blood samples, peripheral blood samples were collected in EDTA tubes from each patient.

### 4.5. Mononuclear Cell (PBMC) Isolation from Peripheral Blood Samples

Blood samples were diluted 1:1 with phosphate-buffered saline (PBS) solution (Capricorn Scientific GmbH, PBS-10X, Ebsdorfergrund, Germany) and layered onto Lymphocyte Separation Medium (LSM, Capricorn Scientific GmbH, Ebsdorfergrund, Germany). Three milliliters of cell separation solution (Histopaque-1077, Sigma-Aldrich, Merck KGaA, Darmstadt, Germany) was added to clean 15 mL Falcon tubes, which were then centrifuged at 2000 rpm for 15 min. The white, cloud-like mononuclear cells remaining between the LSM solution and serum were collected using sterile Pasteur pipettes and transferred to new Falcon tubes. The collected cells were centrifuged at 1200 rpm for five minutes and washed with 7 mL of PBS. The pellet obtained after centrifugation was resuspended in 1 mL of freezing medium containing 10% dimethyl sulfoxide (DMSO, Cat. No. 109678, Merck KGaA, Darmstadt, Germany). and stored at −80 °C until DNA isolation.

### 4.6. DNA Isolation

DNA extraction from mononuclear cells was performed using the High-Purity PCR Template Preparation Kit (Cat. No. 11796828001, Roche Diagnostics GmbH, Mannheim, Germany) according to the manufacturer’s instructions. Extracted DNA samples were stored in Eppendorf tubes at −20 °C until further use.

### 4.7. Comparative Genomic Analysis (CGH)

NimbleGen CGH arrays were hybridized using the NimbleGen Dual-Colour DNA Labelling Kit (Roche NimbleGen, Inc., Madison, WI, USA) according to the instructions provided by the manufacturer and were used for Comparative Genomic Hybridization (CGH) analysis. DNA samples from the five selected cases and control DNA samples were labeled with different fluorochromes and hybridized with metaphase chromosomes extracted from healthy cells. Accordingly, hybridization profiles of the chromosomes were calculated. The method included labeling isolated DNA samples with different fluorochromes (Cy3-green, Cy5-red); treating with Human Cot-1 DNA to reduce nonspecific hybridization; fixing BAC, PAC, cosmid, cDNA, oligonucleotide, and PCR-derived probes on a glass matrix; performing hybridization; and identifying fluorescent signals using an image analysis program. Cytogenetic idiograms of each case were generated by comparative analysis.

### 4.8. Real-Time PCR

The concentrations of the isolated DNA samples were measured, and these samples were evaluated for *DEFA4*, *NDUFV1*, *ADAM6*, *H19*, *ZIM2*, *GNAS*, and *SIX3* genes using real-time RT-PCR. The experiments were performed on a LightCycler^®^ 480 Instrument II (Roche Diagnostics GmbH, Mannheim, Germany) using 2X SYBR Green Master Mix (Procomcure Biotech GmbH, Thalgau, Austria). For the detection of gains or losses in these regions, primer and enzyme mixtures containing labeled probes designed specifically for these regions were used, and the sequences of the primers and probes are listed in Table 4. Relative quantification expression was calculated using the 2^−ΔΔCt^ method, with β-actin as an internal control. For defining copy number alterations, a ΔΔCt threshold of ≥2 was used for amplification, and ≤0.5 for deletion. A healthy reference DNA sample was used as a calibrator.

### 4.9. Evaluation of Hearing Functions of Patients

Audiological assessments were performed before each chemotherapy cycle and at least three months after the completion of cisplatin treatment, with the most recent audiological results used to evaluate ototoxicity. Before cisplatin chemotherapy, all patients underwent routine otorhinolaryngologic examination and audiological evaluation. For children under 3 years of age, hearing thresholds were evaluated via distortion product otoacoustic emissions (DPOAEs) and auditory brainstem responses (ABRs) using 6 and 8 kHz tone burst stimuli. For older children, hearing thresholds were determined using pure tone audiometry and DPOAE tests. Brock and Muenster classifications were used for these ototoxicity evaluations [17].

### 4.10. Statistical Evaluation of Data

Statistical analyses were performed using SPSS 29.0 software (IBM Corp., Armonk, NY, USA). Descriptive and distributional statistics were conducted for all groups. Gene findings that were not normally distributed were compared with cisplatin dose, severity of hearing loss, and gene findings using the non-parametric Kruskal–Wallis test.

The relationship between ∆∆Ct levels of the investigated genes and hearing loss status (based on Brock and Muenster scores) was evaluated using the non-parametric Mann–Whitney U test. Demographic and clinical findings related to patient gender, diagnosis, presence of ototoxicity, and gene status (loss, gain, or normal) were analyzed using the chi-square test. Correlation analyses were performed using the Spearman correlation test. A *p*-value of <0.05 was considered statistically significant for all analyses.

## 5. Conclusions

In conclusion, there are different genetic changes in cases that may or may not show varying levels of hearing problems. If we know these genetic changes, it will be possible to develop patient-specific treatment strategies and thus prevent ototoxicity. Our study identified the *ZIM2* gene as a promising candidate biomarker that warrants further investigation for predicting susceptibility to cisplatin-induced ototoxicity in cancer treatment. In clinical practice, audiologic assessment of patients receiving cisplatin before therapy and at frequent intervals is essential. Patients with *ZIM2* gene amplification may be considered for closer audiological follow-up and, if clinically appropriate, cisplatin dose adjustment or alternative therapies.

## Figures and Tables

**Figure 1 ijms-26-08880-f001:**
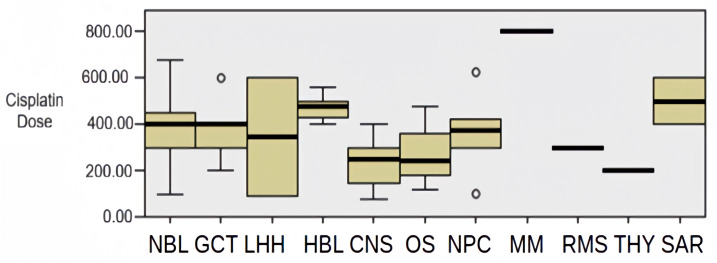
Distribution of cisplatin doses by tumor subtype.

**Table 1 ijms-26-08880-t001:** Distribution of pediatric tumor diagnoses.

Diagnoses	(%)	Patients (n)
Neuroblastoma	42.7	35
Central Nervous System Tumors	14.6	12
Germ Cell Tumors	11	9
Hepatoblastoma	11	9
Nasopharyngeal Carcinoma	6.1	5
Osteosarcoma	4.9	4
Non-Rhabdomyosarcoma(Soft Tissue Sarcoma)	2.4	2
Langerhans Cell Histiocytosis	2.4	2
Thyroid Carcinoma	2.4	2
Rhabdomyosarcoma	1.2	1
Malignant Melanoma	1.2	1

**Table 2 ijms-26-08880-t002:** Properties of the cases with hearing loss.

Case No.	Sex	Age (Months)	Diagnosis	Cisplatindose (mg/m^2^)	Hearing Loss	Ototoxicity *	Brock	Muenster	Gene Mutation
CO33	M	13	Neuroblastoma	300	Severe	Positive	GRADE 2	grade 3a	*H19*del, *GNAS*del
CO19	F	20	Neuroblastoma	100	Mild	Negative	GRADE 0	grade 2a	*ADAM6*amp, *H19*amp, *SIX3*amp
CO42	F	16	Neuroblastoma	300	Mild	Negative	GRADE 0	grade 2a	*DEFA4*del, *NDUFV1*del, *ADAM6*del
*CO27*	M	21	Neuroblastoma	600	Severe	Positive	GRADE 1	grade 2c	*NDUFV1*amp, *ADAM6*amp, *SIX3*amp
*CO41*	M	51	Neuroblastoma	300	Severe	Positive	GRADE 1	grade 2c	*ADAM6*amp, *SIX3*amp
*CO46*	M	48	Neuroblastoma	600	Severe	Positive	GRADE 1	grade 2b	*NDUFV1*amp, *ADAM6*amp, *SIX3*amp
*CO30*	M	35	Neuroblastoma	450	Severe	Positive	GRADE 2	grade 3c	*GNAS*amp, *SIX3*amp
YCO2	F	76	Neuroblastoma	300	Mild	Positive	GRADE 2	grade 1	*ZIM2*del, *H19*del
*CO28*	M	29	Neuroblastoma	300	Severe	Positive	GRADE 1	grade 2b	*DEFA4*amp
CO53	F	2	Neuroblastoma	400	Mild	Negative	GRADE 0	grade 2a	*H19*del
*CO34*	F	31	Neuroblastoma	300	Severe	Positive	GRADE 2	grade 2b	*SIX3*del
*CO18*	M	18	Neuroblastoma	300	Severe	Positive	GRADE 2	grade 2c	*ZIM2*amp
*CO36*	M	71	Neuroblastoma	600	Severe	Positive	GRADE 1	grade 2b	*H19*del
CO21	M	23	Neuroblastoma	675	Mild	Negative	GRADE 1	grade 2a	-
CO12	F	20	Neuroblastoma	100	Mild	Negative	GRADE 0	grade 2a	-
*CO20*	F	50	Neuroblastoma	450	Severe	Positive	GRADE 2	grade 2c	-
CO9	M	60	Neuroblastoma	450	Mild	Positive	GRADE 0	grade 2a	-
*CO11*	M	27	Hepatoblastoma	560	Severe	Positive	GRADE 2	grade 2b	*DEFA4*amp
CO44	F	18	Hepatoblastoma	480	Mild	Negative	GRADE 0	grade 2a	*SIX3*del
*CO54*	M	12	Hepatoblastoma	400	Severe	Positive	GRADE 2	grade 2c	*ZIM2*amp
CO97	M	23	Hepatoblastoma	400	Mild	Positive	GRADE 1	grade 2a	*DEFA4*amp
*CO40*	M	83	Hepatoblastoma	430	Severe	Positive	GRADE 1	grade 2b	*DEFA4*del, *H19*del, *GNAS*del, *SIX3d*el
*YCO1*	M	134	CNS Tumor	120	Severe	Positive	GRADE 3	grade 2c	*ZIM2*amp
*YCO9*	M	68	CNS Tumor	300	Severe	Positive	GRADE 3	grade 3b	*GNAS*amp, *SIX3*del
CO65	F	100	CNS Tumor	75	Mild	Positive	GRADE 0	grade 1	*NDUFV1*amp, *H19*amp, *SIX3* amp
YCO4	F	169	CNS Tumor	175	Mild	Positive	GRADE 2	grade 2a	*ZIM2*del, *NDUFV1*amp
*CO39*	M	152	Nasopharynx CA	425	Severe	Positive	GRADE 4	grade 2a	*ZIM2*amp, *NDUFV1*del, *H19*del
*C037*	M	149	Nasopharynx CA	300	Severe	Positive	GRADE 1	grade 2c	DEFA4amp
*CO63*	M	174	Nasopharynx CA	375	Severe	Positive	GRADE 1	grade 3a	*H19*del, *SIX3*del
*CO35*	F	75	Osteosarcoma	240	Severe	Positive	GRADE 1	grade 2b	*DEFA4*amp, *GNAS*amp
CO62	M	187	Germ Cell Tumor	400	Mild	Negative	GRADE 0	grade 2a	*ZIM2*del

F: female, M: male. * Ototoxicity was evaluated according to both clinical features and Brock and Muenster grades.

**Table 3 ijms-26-08880-t003:** Distribution of mutations in cases with ototoxicity and hearing loss compared to all cases.

	Mutation	*ADAM6*	*SIX3*	*GNAS*	*NDUFV1*	*H19*	*DEFA4*	*ZIM2*
All cases (N: 82)	Amplification Deletion	15 2	16 18	8 12	13 3	7 24	12 2	10 11
Ototoxicity (N: 24)	Amplification Deletion	3 -	6 3	3 2	4 1	1 6	5 1	4 2
Severe hearing loss (N: 19)	Amplification Deletion	3 -	5 3	3 2	2 1	- 5	4 1	4 -
Mild hearing loss (N: 12)	Amplification Deletion	1 1	2 1	- -	2 1	2 2	1 1	- 3

**Table 4 ijms-26-08880-t004:** Primer and probe sequences used for real-time RT-PCR analysis.

Gene	Forward Primer (5′→3′) Reverse Primer (5′→3′) Probe (5′→3′)
*DEFA4*	5′-CCAGGCAAGAGGTGATGAG-3′
	5′-TGAAACCTGAAGAAGCAGAGC-3′
	5′-FAM-TGGGCCAGAAGACCAGGACATATCTA-BBQ-3′
*NDUFV1*	5′-GATCTTACGCCATGATCCTCAC-3′
	5′-GAGGCCTCATTGTAGAATTCCC-3′
	5′-FAM-ATGTAGATATAGGCAGCGCGGGC-BBQ-3′
*ADAM6*	5′-CTGTCCTACAGCCTGTGTTT-3′
	5′-CTGAGTTGTCACCAGCAGAT-3′
	5′-FAM-TCACATGCGGAGGAAACACCTTCT-BBQ-3′
*H19*	5′-ACGTGTCGCTATCTCTAGGT-3′
	5′-GCTGTTCCGATGGTGTCTT-3′
	5′-FAM-AACCAGACCTCATCAGCCCAACAT-BBQ-3′
*ZIM2*	5′-CCCTTTGAATGTGGTAGTGAGA-3′
	5′-TTGGCTGTGACTCGGTAAAG-3′
	5′-FAM-AAAGCCATGAGCGTGAGCAGC-BBQ-3′
*GNAS*	5′-CGAAGGTGCGTTACCAGATT-3′
	5′-TTTCAGCACGGGTAGAGTTAA-3′
	5′-FAM-TTTGTGCTGGGTCATCAGAGCAGA-BBQ-3′
*SIX3*	5′-CCTCTTCCTCCTCTTCCTTCT-3′
	5′-GAGTGTGTGTGGACTTGTATGT-3′
	5′-FAM-ACAAACCGAAATCAGGATACCCAACCA-BBQ-3′

## Data Availability

The data that support the findings of this study are available from the corresponding author upon reasonable request.

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
