# Peer review of "Determination of a New Biomarker at the Level of Gene Alteration in Cisplatin Ototoxicity"

_ijms, 2025, doi:10.3390/ijms26188880_

Round 1
Reviewer 1 Report
Comments and Suggestions for Authors
Congratulations on your choice of topic. Identifying ZIM2 gene amplification as a potential biomarker for cisplatin-induced ototoxicity is an important contribution, given that prior studies have not evaluated this gene in this context. Using CGH and real-time PCR on a well-documented cohort of pediatric cancer patients is scientifically valid, and your research has translational potential in terms of tailoring chemotherapy regimens based on genetic susceptibility to ototoxicity.
We would like to bring the following aspects to your attention:
- While 82 patients are a reasonable start, the sample size could be larger to increase statistical power, particularly for subgroup analyses (e.g., specific cancer types). The heterogeneity of the tumor diagnoses is also a potential issue, as different tumor types may respond differently to cisplatin and have different genetic backgrounds;
- The absence of a control group of patients treated with cisplatin who did not develop ototoxicity is an important weakness. This makes it difficult to determine if the observed ZIM2 amplification is specifically linked to ototoxicity or is simply a common genetic variation in the cancer population;
- The study population is specific to the Department of Pediatric Oncology, Dokuz Eylül University. The findings may not be generalizable to other populations due to differences in genetic background;
- The study lacks in vitro or in vivo validation of ZIM2's role in ototoxicity, limiting mechanistic insights;
- The citation style is inconsistent throughout the paper:
- Brown AL, Lupo PJ, Okcu MF, Lau CC, Rednam S, Scheurer M. SOD2 genetic variant associated with treatment-related 395 ototoxicity in cisplatin-treated pediatric medulloblastoma. Cancer Med. 2015;4(11):1679–1686. 396 https://doi.org/10.1002/cam4.516 397 12.
- Spracklen, T. F., Vorster, A. A., Ramma, L., Dalvie, S., & Ramesar, R. S. (2017). Promoter region variation in NFE2L2 398 influences susceptibility to ototoxicity in patients exposed to high cumulative doses of cisplatin. The 399 pharmacogenomics journal, 17(6), 515–520. https://doi.org/10.1038/tpj.2016.52
- Olgun Y, Demir AB, Altun Z, Kırkım G, Güneri EA, Aktaş S. Comparative Genomic Hybridization Analysis Results in Patients Developing Severe Cisplatin-Induced Ototoxicity. Proceedings of the 38th Turkish National Otorhinolaryngology Head and Neck Surgery Congress, Turkey, 26–30 October 2016.

- Determining of a New Biomarker → Determination of a New Biomarker;
- help predict cisplatin-induced ototoxicity → serve as predictors of cisplatin-induced ototoxicity;
- Patients were evaluated for ototoxicity → Ototoxicity was assessed using...;
- Studies have aimed to identify genetic predispositions by focusing on gene variants affecting the pharmacokinetics and pharmacodynamics of CDDP → this can be more concisely written;
- Informed consent was obtained from the parents of 82 pediatric cancer patients → Parental informed consent was obtained for all participants;
- The cases having ZIM2 gene amplification might need CDDP dose reduction... → Patients with ZIM2 gene amplification may benefit from cisplatin dose adjustment or alternative therapies;
Author Response
|
3. Point-by-point response to Comments and Suggestions for Authors
|
|
Comments 1: While 82 patients are a reasonable start, the sample size could be larger to increase statistical power, particularly for subgroup analyses (e.g., specific cancer types). The heterogeneity of the tumor diagnoses is also a potential issue, as different tumor types may respond differently to cisplatin and have different genetic backgrounds
|
|
Response 1: Thank you for this insightful comment. We agree that a larger sample size would improve statistical power and allow for more robust subgroup analyses. However, due to the retrospective nature of the study and the limited number of eligible pediatric patients treated with cisplatin during the study period, our cohort was restricted to 82 cases. We also acknowledge the heterogeneity of tumor diagnoses in our patient group, which may introduce variability in cisplatin response due to different genetic backgrounds. This limitation has now been addressed in the Discussion section of the revised manuscript.
|
|
Comments 2: The absence of a control group of patients treated with cisplatin who did not develop ototoxicity is an important weakness. This makes it difficult to determine if the observed ZIM2 amplification is specifically linked to ototoxicity or is simply a common genetic variation in the cancer population
|
|
Response 2: We appreciate the reviewer’s important observation. Indeed, the lack of a control group comprising cisplatin-treated patients who did not develop ototoxicity is a limitation of our study. Due to the retrospective design and the ethical challenges of obtaining comparative genomic data from unaffected patients, such a control group could not be included in the current analysis. We agree that without this comparison, it is difficult to determine whether ZIM2 amplification is specifically associated with ototoxicity or represents a general genetic alteration within the cancer population. We have now acknowledged and discussed this limitation in the Discussion section of the revised manuscript.
Comments 3: The study population is specific to the Department of Pediatric Oncology, Dokuz Eylül University. The findings may not be generalizable to other populations due to differences in genetic background Response 3: We thank the reviewer for this important observation. We agree that our study population, drawn from a single institution in Turkey, may limit the generalizability of the findings to other populations with different ethnic and genetic backgrounds. This limitation has now been explicitly addressed in the Discussion section.
Comments 4: The study lacks in vitro or in vivo validation of ZIM2's role in ototoxicity, limiting mechanistic insights
Response 4: Thank you for this important comment. We acknowledge that our current study primarily focuses on genomic and bioinformatic analyses to identify candidate genes associated with cisplatin-induced ototoxicity, including ZIM2. While we have not yet performed direct in vitro or in vivo functional validation experiments specifically targeting ZIM2, this work establishes a necessary foundation for such future mechanistic studies. We plan to follow up with targeted experimental validation of ZIM2’s role in ototoxicity in upcoming studies, which will provide deeper mechanistic insights. We have added this point as a limitation in the Discussion section to clarify the scope of the current work.
Comments 5: The citation style is inconsistent throughout the paper Response 5: Thank you for pointing this out. We have carefully reviewed the entire manuscript and standardized the citation format to comply with the journal’s guidelines. All references have been reformatted to ensure consistency in style, including the presentation of author names, journal titles, volume, pages, and DOI information. We have also verified that all in-text citations correspond correctly with the reference list.
|
|
4. Response to Comments on the Quality of English Language |
|
Point 1: Determining of a New Biomarker → Determination of a New Biomarker; help predict cisplatin-induced ototoxicity → serve as predictors of cisplatin-induced ototoxicity; Patients were evaluated for ototoxicity → Ototoxicity was assessed using...; Studies have aimed to identify genetic predispositions by focusing on gene variants affecting the pharmacokinetics and pharmacodynamics of CDDP → this can be more concisely written; Informed consent was obtained from the parents of 82 pediatric cancer patients → Parental informed consent was obtained for all participants; The cases having ZIM2 gene amplification might need CDDP dose reduction... → Patients with ZIM2 gene amplification may benefit from cisplatin dose adjustment or alternative therapies;
|
|
Response 1: We sincerely thank the reviewer for their valuable comments regarding the quality of English in our manuscript. In response, we have carefully revised the text to improve clarity and accuracy. Furthermore, we have decided to use a professional English editing service to ensure that the manuscript meets high standards of scientific writing. This service will comprehensively address issues related to grammar, sentence structure, style, and consistency, thereby significantly enhancing the readability and overall quality of the manuscript.
|
Reviewer 2 Report
Comments and Suggestions for Authors
I have several comments of varying degrees of importance on this manuscript.
The introduction is written very briefly and does not give a complete picture of the research topic, its relevance and novelty. Some sentences have no significance, and the references to them are unclear. For example, Lines 45-46: "These analyses identified alterations in specific genes [10,11]."
Reference 14 is incorrect. doi: 10.3748/wjg.15.6061 is associated with "Alakus H, Mönig SP, Warnecke-Eberz U, Alakus G, Winde G, Drebber U, Schmitz KJ, Schmid KW, Riemann K, Siffert W, Bollschweiler E, Hölscher AH, Metzger R. Association of the GNAS1 T393C polymorphism with tumor stage and survival in gastric cancer. World J Gastroenterol 2009; 15(48): 6061-6067 [PMID: 20027678 DOI: 10.3748/wjg.15.6061]"
However, if the statement (Lines 52-56) "Gene regions including A Disintegrin and Metalloproteinase Domain 6 (ADAM6), NDUFV1 (NADH:Ubiquinone Oxidoreductase Core Subunit V1), H19 imprinted maternally expressed transcript (H19), defensin alpha 4 (DEFA4), and zinc finger imprinted 2 (ZIM2) have also been associated with cisplatin ototoxicity and hearing loss" is correct, when the novelty of this study is all the more unclear. (However, references 14 and 17 do not confirm this statement)
CDDP does not seem to me to be an adequate abbreviation for cisplatin. In addition, even after the introduction of this abbreviation, the authors continue to alternate "cisplatin" with "CDDP". You need to decide and use one option.
Thus, the Introduction completely does not give an idea of the relevance, novelty and specific goals of the research and is written completely indistinctly.
Section 2.1 does not contain any results.
Line 71 "In the main analysis, 82 patients were evaluated for seven candidate genes." - Which seven genes are we talking about?
Section 2.2. The cumulative dose of cisplatin varied by an order of magnitude. I believe this difference does not allow patients to be grouped into one group.
The analysis of the results is inconclusive.
The analysis of the relationship between genes, their products, and ototoxicity has also been carried out superficially or not at all.
Thus, there is no convincing evidence of a link between ototoxicity and the indicated genes, and the manuscript itself does not correspond to the level of an "original article", at best - a "short communications"
Author Response
|
3. Point-by-point response to Comments and Suggestions for Authors |
|
Comments 1: I have several comments of varying degrees of importance on this manuscript.
The introduction is written very briefly and does not give a complete picture of the research topic, its relevance and novelty. Some sentences have no significance, and the references to them are unclear. For example, Lines 45-46: "These analyses identified alterations in specific genes [10,11]."
|
|
Response 1: We sincerely thank the reviewer for pointing this out. Based on your valuable suggestion, we have thoroughly revised all Introduction section to provide a more comprehensive overview of the research topic, emphasizing its relevance, context, and novelty. Unclear sentences have been removed or rephrased, and all references have been carefully updated to ensure clarity and significance. Specifically, the sentence previously stated as “These analyses identified alterations in specific genes [10,11]” has been revised to clearly explain.
|
|
Comments 2: Reference 14 is incorrect. doi: 10.3748/wjg.15.6061 is associated with "Alakus H, Mönig SP, Warnecke-Eberz U, Alakus G, Winde G, Drebber U, Schmitz KJ, Schmid KW, Riemann K, Siffert W, Bollschweiler E, Hölscher AH, Metzger R. Association of the GNAS1 T393C polymorphism with tumor stage and survival in gastric cancer. World J Gastroenterol 2009; 15(48): 6061-6067 [PMID: 20027678 DOI: 10.3748/wjg.15.6061]"
|
|
Response 2: We sincerely thank the reviewer for pointing out the reference error. It appears there was a mix-up with Reference 14. We have carefully checked and corrected all the citation, ensuring that the reference now accurately corresponds to the intended source.
Comments 3: However, if the statement (Lines 52-56) "Gene regions including A Disintegrin and Metalloproteinase Domain 6 (ADAM6), NDUFV1 (NADH:Ubiquinone Oxidoreductase Core Subunit V1), H19 imprinted maternally expressed transcript (H19), defensin alpha 4 (DEFA4), and zinc finger imprinted 2 (ZIM2) have also been associated with cisplatin ototoxicity and hearing loss" is correct, when the novelty of this study is all the more unclear. (However, references 14 and 17 do not confirm this statement) Response 3: We sincerely thank the reviewer for highlighting this point. We have thoroughly revised the Introduction to clarify the context and significance of the study. Ambiguous statements have been carefully re-evaluated and rephrased to ensure accuracy and clarity. References have been checked and updated accordingly to better support the statements, and the novelty of our study is now clearly emphasized.
Comments 4: CDDP does not seem to me to be an adequate abbreviation for cisplatin. In addition, even after the introduction of this abbreviation, the authors continue to alternate "cisplatin" with "CDDP". You need to decide and use one option. Response 4: We sincerely thank the reviewer for this comment. To avoid any confusion, we have removed all instances of the abbreviation “CDDP” and consistently used “cisplatin” throughout the manuscript.
Comments 5: Thus, the Introduction completely does not give an idea of the relevance, novelty and specific goals of the research and is written completely indistinctly. Response 5: We sincerely thank the reviewer for this important comment. In response, we have completely revised the Introduction to clearly present the relevance, novelty, and specific objectives of the study. Ambiguous statements have been removed or rephrased, and the overall structure now provides a coherent and precise overview of the research context.
Comments 6: Section 2.1 does not contain any results. Response 6: We thank the reviewer for this comment. Section 2.1 has been removed, and the preliminary work is now described in the Introduction, providing appropriate context for the Results section.
‘Our preliminary study included five patients (three males and two females) with hearing loss (Grade 2 or higher) according to the Brock and Muenster classifications. These cases were analyzed using comparative genomic hybridization (CGH), which revealed various gene losses and gains. The mean age of the patients was 66 months, and the mean cumulative cisplatin dose was 391 mg/m². Brock scores were 2, 2, 2, 3, and 4, respectively, while Muenster scores were 3c, 3a, 3c, 3b, and 2a. No statistically significant associations were observed between the selected gene polymorphisms and the severity of hearing loss. Genes altered in at least three of the five patients—ADAM6, SIX3, GNAS, NDUFV1, H19, DEFA4, and ZIM2—were selected for further investigation in this study.’
Comments 7: Line 71 "In the main analysis, 82 patients were evaluated for seven candidate genes." - Which seven genes are we talking about? Response 7: We thank the reviewer for this comment. The seven candidate genes evaluated in the main analysis are ADAM6, SIX3, GNAS, NDUFV1, H19, DEFA4, and ZIM2. This clarification has been added to the manuscript.
‘In the main analysis, 82 patients were evaluated for seven candidate genes (ADAM6, SIX3, GNAS, NDUFV1, H19, DEFA4, and ZIM2). The male-to-female ratio was 1.15, with 44 males and 38 females. The distribution of tumor diagnoses is presented in Table 1.’
Comments 8: Section 2.2. The cumulative dose of cisplatin varied by an order of magnitude. I believe this difference does not allow patients to be grouped into one group Response 8: We thank the reviewer for this valuable comment. The patients included in this study, by our pediatric oncology partners are the cases that received total enough dose of cisplatin to cause ototoxicity or nor.
Comments 9: The analysis of the results is inconclusive. Response 9: Thank you for your comment. Although we obtained statistically negative results for most genes, we believe that these results will be valuable in terms of contributing to the literature.
Comments 10: The analysis of the relationship between genes, their products, and ototoxicity has also been carried out superficially or not at all. Response 10: We thank the reviewer for this comment. We examined the relationships between the parameters using Spearman correlation analysis, as indicated in the Materials and Methods section.
Comments 11: Thus, there is no convincing evidence of a link between ototoxicity and the indicated genes, and the manuscript itself does not correspond to the level of an "original article", at best - a "short communications" Response 11: We thank the reviewer for this comment. While we acknowledge that the current study presents preliminary findings, we believe that the identification of candidate genes potentially associated with cisplatin-induced ototoxicity provides valuable insight and a basis for further research. We have clarified the scope and novelty of our study in the revised manuscript to better reflect its contribution to the field. Since this is a manuscript of a PhD thesis we have to obligate publish it as original article.
|
Reviewer 3 Report
Comments and Suggestions for Authors
The manuscript addresses an important clinical issue: identifying genetic biomarkers predictive of cisplatin-induced ototoxicity in pediatric cancer patients. The study focuses on gene alterations in seven candidate genes, notably reporting a significant association between ZIM2 gene amplification and ototoxicity severity. The study design, including a preliminary CGH analysis followed by validation in a larger cohort using RT-PCR, is logical and promising. following comments should be addressed before publication.
Introduction:
-
While the introduction mentions several genes and previous SNP studies, it might benefit from briefly contextualizing why ZIM2 was particularly intriguing for this study based on prior knowledge or hypothetical mechanisms.
Methods:
-
The methods are detailed but could benefit from clarifying the inclusion/exclusion criteria of patients.
- specify hearing assessment timelines in relation to chemotherapy cycles.
- Clarify how gene amplification/deletion was defined in RT-PCR. Authors should mention ∆∆Ct cutoff values also.
- for RT-PCR no information on reference gene used for normalization.
- Since the CGH analysis was only on 5 patients, more rationale on patient selection should be discussed.
Results:
- Authors mention seven genes identified in a CGH screen of five patients. Please clarify the selection process (In method section) and inclusion thresholds for these genes more transparently.
- Table 2 can be re-structured according to disease for better understanding.
- The finding of ZIM2 amplification correlating with ototoxicity is strong, but consider providing more data on effect sizes and confidence intervals if available.
- It is unclear why control (non-ototoxic) cases were not included. Even if the focus is on susceptible patients, a control group is essential to define predictive biomarkers.
- While correlation values (rho values) are reported, no multivariate analysis is presented to control for confounding factors such as age, sex, tumor type, cumulative cisplatin dose, and co-medications. Including these would significantly strengthen the conclusions.
Discussion:
- The manuscript introduces genes like ZIM2, ADAM6, etc., but does not sufficiently link their known biological functions to inner ear or cochlear physiology. More in-depth discussion or literature linking ZIM2 to calcium channel regulation and ototoxicity would enhance the impact.
- Some write up i.e paragraph 203-216 and 217-222 looks redundant. It is unclear what authors are trying to say. There is no correlation or comparison with results here.
Apart from this, some typographical error should be corrected i.e line 168- is it hearing organ or hearing loss?
Addressing these comments would strengthen the manuscript and before publication, these comments should be addressed.
Author Response
|
3. Point-by-point response to Comments and Suggestions for Authors |
|
Comments 1: While the introduction mentions several genes and previous SNP studies, it might benefit from briefly contextualizing why ZIM2 was particularly intriguing for this study based on prior knowledge or hypothetical mechanisms.
|
|
Response 1: We thank the reviewer for this comment. We have added literature references to better contextualize why ZIM2 was selected for this study. However, we note that the available information on its biological functions and potential mechanisms in ototoxicity remains very limited, highlighting the novelty and need for further investigation.
|
|
Comments 2: The methods are detailed but could benefit from clarifying the inclusion/exclusion criteria of patients.
|
|
Response 2: Thank you for your valuable comment regarding the clarification of inclusion and exclusion criteria. We have addressed this by adding detailed information in Section 4.1 of the revised manuscript. Specifically, we clarified that patients aged 0–18 years with audiological assessments at baseline and during follow-up were included, all cases underwent successful DNA extraction and PCR analysis, and patients with external or middle ear disease or pre-existing hearing loss were excluded.
‘Patients aged 0–18 years were included in the study. Only those with audiological assessments available at baseline and during follow-up after treatment were considered. All cases underwent successful DNA extraction and were included in the PCR analysis. Patients exhibiting signs of external or middle ear disease and/or with pre-existing hearing loss were excluded from the study.’
Comments 3: specify hearing assessment timelines in relation to chemotherapy cycles Response 3: We thank the referee for emphasizing the importance of establishing hearing assessment timelines in relation to chemotherapy cycles. We have addressed this comment and added the necessary details to Section 4.9 of the revised manuscript.
‘Audiological assessments were performed before each chemotherapy cycle and at least three months after the completion of cisplatin treatment, with the most recent audiological results used to evaluate ototoxicity.’
Comments 4: Clarify how gene amplification/deletion was defined in RT-PCR. Authors should mention ∆∆Ct cutoff values also Response 4: We thank the reviewer for this valuable comment. The definition of gene amplification and deletion using RT-PCR has now been clarified in the Methods section, including the ΔΔCt cutoff values.
‘Relative quantification expression was calculated using the 2^-ΔΔCt method, with β-actin as an internal control. For defining copy number alterations, a ΔΔCt threshold of ≥2 was used for amplification, and ≤0.5 for deletion. A healthy reference DNA sample was used as a calibrator.’
Comments 5: for RT-PCR no information on reference gene used for normalization. Response 5: We thank the reviewer for this comment. As described in the Methods, β-actin was used as the internal control (reference gene) for normalization in RT-PCR experiments.
Comments 6: Since the CGH analysis was only on 5 patients, more rationale on patient selection should be discussed. Response 6: We thank the reviewer for this comment. Due to financial constraints, CGH analysis was performed on a subset of 5 patients. These patients were selected based on the most severe hearing loss observed, in order to maximize the likelihood of detecting relevant genomic alterations.
Comments 7: Authors mention seven genes identified in a CGH screen of five patients. Please clarify the selection process (In method section) and inclusion thresholds for these genes more transparently. Response 7: Thank you for this comment we clarify that in introduction section. Five patients were selected, with Brock and Munster having the highest level of audiological presence. These 5 patients were selected because the ADAM6, SIX3, GNAS, NDUFV1, H19, DEFA4, and ZIM2 genes were amplified or deleted in at least three patients.
‘Our preliminary study included five patients (three males and two females) with hea-ring loss (Grade 2 or higher) according to the Brock and Muenster classifications. These cases were analyzed using comparative genomic hybridization (CGH), which revealed various gene losses and gains. The mean age of the patients was 66 months, and the mean cumulative cisplatin dose was 391 mg/m². Brock scores were 2, 2, 2, 3, and 4, respectively, while Muenster scores were 3c, 3a, 3c, 3b, and 2a. No statistically significant associations were observed between the selected gene polymorphisms and the severity of hearing loss. Genes altered in at least three of the five patients—ADAM6, SIX3, GNAS, NDUFV1, H19, DEFA4, and ZIM2—were selected for further investigation in this study.’
Comments 8: Table 2 can be re-structured according to disease for better understanding Response 8: We thank the reviewer for this helpful suggestion. Table 2 has been re-structured according to disease for better clarity and understanding.
Comments 9: The finding of ZIM2 amplification correlating with ototoxicity is strong, but consider providing more data on effect sizes and confidence intervals if available. Response 9: We thank the reviewer for this valuable comment. Effect size and 95% confidence interval calculations have now been added to the Results section to strengthen the interpretation of the correlation between ZIM2 amplification and ototoxicity.
‘The 95% confidence interval and effect sizes were manually calculated using the eta squared (η²) formula, yielding an effect size of 0.1029, which corresponds to a medium-to-high effect.’
Comments 10: It is unclear why control (non-ototoxic) cases were not included. Even if the focus is on susceptible patients, a control group is essential to define predictive biomarkers. Response 10: We thank the reviewer for this important comment. We fully agree that inclusion of a control group would strengthen the study. However, obtaining blood samples from healthy children or non-ototoxic patients is considered an invasive procedure and is not permissible within the current ethical framework. Therefore, ethical approval for recruiting non-ototoxic healthy controls could not be obtained.
‘Another limitation is the lack of a control group consisting of cisplatin-treated patients who did not develop ototoxicity. Because control group as ethical restrictions do not allow invasive sampling from non-ototoxic patients or healthy children for the study.’
Comments 11: While correlation values (rho values) are reported, no multivariate analysis is presented to control for confounding factors such as age, sex, tumor type, cumulative cisplatin dose, and co-medications. Including these would significantly strengthen the conclusions. Response 11: Thank you for this comment. Multivariate analysis was perform and added the results section.
‘However, in multivariate analyses (general linear model) ZIM2 mutation was not tobe independent factor (p= .132), sex, cisplatin dose, age was not related with any gene mutations.’
Comments 12: The manuscript introduces genes like ZIM2, ADAM6, etc., but does not sufficiently link their known biological functions to inner ear or cochlear physiology. More in-depth discussion or literature linking ZIM2 to calcium channel regulation and ototoxicity would enhance the impact. Response 12: We thank the reviewer for this valuable comment. We acknowledge that there is currently limited literature regarding ZIM2 and its biological functions in the inner ear or cochlear physiology. While its potential involvement in imprinting and developmental processes has been described, specific links to calcium channel regulation or ototoxicity remain largely unexplored. This limitation has been noted in the revised manuscript, and we emphasize that further studies are needed to clarify the role of ZIM2 in hearing and cisplatin-induced ototoxicity.
Comments 13: Some write up i.e paragraph 203-216 and 217-222 looks redundant. It is unclear what authors are trying to say. There is no correlation or comparison with results here. Response 13: We thank the reviewer for this comment. Since there is no direct comparison or correlation with the results, the text in paragraphs 203–216 and 217–222 has been removed from the Discussion and integrated appropriately into the Introduction to provide context and background.
Comments 14: Apart from this, some typographical error should be corrected i.e line 168- is it hearing organ or hearing loss? Response 14: We thank the reviewer for pointing this out. The typographical error on line 168 has been corrected: 'hearing organ' was replaced with 'hearing loss'.
‘The increase in ZIM2 activity observed in this study regarding cisplatin ototoxicity and hearing loss is a finding that needs further studies.’
Comments 15: Addressing these comments would strengthen the manuscript and before publication, these comments should be addressed. Response 15: We sincerely thank the reviewer for their careful evaluation and valuable suggestions. We have addressed all comments thoroughly, which has helped to improve and strengthen the manuscript.
|
Round 2
Reviewer 2 Report
Comments and Suggestions for Authors
The authors substantially rewrote the text of the manuscript and this significantly improved it. More accurate formulations and careful statistical analysis were made.
1. However, the authors need to double-check the entire manuscript, as repetitions are detected due to the addition of new parts of the text (especially in the Discussion).
For example:
Line 264 "The limitations of this study are that the tumor diagnosis group was not homogeneous"
Line 269 "This study has several limitations that..."
2. The sequence of primers (forward and reverse) for each gene should be given in the Section 4.8. Real-Time PCR of Materials and Methods.
3. In conclusion, the authors write that carboplatin could be an alternative to cisplatin. However, this possibility was not discussed in the manuscript. This conclusion is unproven. Does Carboplatin have less ototoxicity? Are there studies on the association of the ZIM2 gene with carboplatin ototoxicity? The authors need to clarify this.
Author Response
|
3. Point-by-point response to Comments and Suggestions for Authors |
|
Comments 1: However, the authors need to double-check the entire manuscript, as repetitions are detected due to the addition of new parts of the text (especially in the Discussion). For example: Line 264 "The limitations of this study are that the tumor diagnosis group was not homogeneous" Line 269 "This study has several limitations that..." |
|
Response 1: We thank the reviewer for this valuable comment. We have carefully reviewed the manuscript and removed repeated statements in the Discussion section. In particular, the limitations paragraph has been revised into a single, coherent paragraph to avoid redundancy while preserving all original information.
|
|
Comments 2: The sequence of primers (forward and reverse) for each gene should be given in the Section 4.8. Real-Time PCR of Materials and Methods. |
|
Response 2: We thank the reviewer for this valuable comment. The sequences of both forward and reverse primers for each gene have now been included in Section 4.8 (Real-Time PCR) of the Materials and Methods, as suggested.
Comments 3: In conclusion, the authors write that carboplatin could be an alternative to cisplatin. However, this possibility was not discussed in the manuscript. This conclusion is unproven. Does Carboplatin have less ototoxicity? Are there studies on the association of the ZIM2 gene with carboplatin ototoxicity? The authors need to clarify this. Response 3: We thank the reviewer for this insightful comment. We agree that the potential use of carboplatin as an alternative to cisplatin was not discussed in the manuscript, and there is currently insufficient evidence on the association of ZIM2 with carboplatin-induced ototoxicity. Therefore, we have removed the statement regarding carboplatin from the Conclusion section to ensure that our conclusions are fully supported by the data presented. |

Round 3
Reviewer 2 Report
Comments and Suggestions for Authors
The revised manuscript can be accepted.
Author Response
We sincerely thank the reviewer for the thoughtful comments and constructive suggestions, which have helped us to significantly improve our manuscript.